# Plasma membrane H⁺-ATPase regulation is required for auxin gradient formation preceding phototropic growth

Tim Hohm[1,2,†], Emilie Demarsy[3,†], Clément Quan[3], Laure Allenbach Petrolati[3], Tobias Preuten[3], Teva Vernoux[4], Sven Bergmann[1,2,*,††] & Christian Fankhauser[3,**,††]

## Abstract

Phototropism is a growth response allowing plants to align their photosynthetic organs toward incoming light and thereby to optimize photosynthetic activity. Formation of a lateral gradient of the phytohormone auxin is a key step to trigger asymmetric growth of the shoot leading to phototropic reorientation. To identify important regulators of auxin gradient formation, we developed an auxin flux model that enabled us to test *in silico* the impact of different morphological and biophysical parameters on gradient formation, including the contribution of the extracellular space (cell wall) or apoplast. Our model indicates that cell size, cell distributions, and apoplast thickness are all important factors affecting gradient formation. Among all tested variables, regulation of apoplastic pH was the most important to enable the formation of a lateral auxin gradient. To test this prediction, we interfered with the activity of plasma membrane H⁺-ATPases that are required to control apoplastic pH. Our results show that H⁺-ATPases are indeed important for the establishment of a lateral auxin gradient and phototropism. Moreover, we show that during phototropism, H⁺-ATPase activity is regulated by the phototropin photoreceptors, providing a mechanism by which light influences apoplastic pH.

**Keywords** auxin; modeling; phototropins; phototropism; plasma membrane H⁺-ATPase

**Subject Categories** Plant Biology; Quantitative Biology & Dynamical Systems

**Mol Syst Biol. (2014) 10: 751**

## Introduction

The ability of plants to adjust their growth to the direction of incoming light already intrigued Greek philosophers in ancient times (Whippo & Hangarter, 2006) and lost nothing of its fascination today. Several key steps of this process termed phototropism starting with light perception and leading to directional growth are well understood (Sakai & Haga, 2012; Christie & Murphy, 2013; Hohm *et al*, 2013). The formation of a lateral gradient of the phytohormone auxin across a unilaterally irradiated hypocotyl (embryonic stem) is necessary and sufficient to cause asymmetric growth and subsequent phototropic bending (Baskin *et al*, 1986; Friml *et al*, 2002; Haga & Sakai, 2012). Yet, the mechanisms for forming this gradient remain elusive. Several auxin carriers including members of the PIN-FORMED (PIN), ATP-binding cassette transporters subfamily B (ABCB), and AUXIN-RESISTANT1 (AUX1) families have been implicated in auxin gradient formation (Friml *et al*, 2002; Blakeslee *et al*, 2004; Stone *et al*, 2008; Christie *et al*, 2011; Ding *et al*, 2011; Willige *et al*, 2013). Light perception by the photoreceptor phototropin1 (phot1) leads to inhibition of ABCB19 activity, which controls the basipetal flux of auxin in the hypocotyl and thereby indirectly modulates lateral auxin gradient formation (Christie *et al*, 2011). The auxin transporters that have most prominently been implicated in formation of a lateral auxin gradient are members of the PIN family (Ding *et al*, 2011; Haga & Sakai, 2012; Willige *et al*, 2013). However, how light affects PIN activity and the importance of intracellular localization of PIN proteins upon phototropic stimuli are still a matter of debate (Christie *et al*, 2011; Ding *et al*, 2011; Sakai & Haga, 2012; Hohm *et al*, 2013).

In addition to being actively transported, protonated auxin is able to diffuse across the plasma membrane (Krupinski & Jonsson, 2010). The protonated fraction of the weak acid auxin [pKₐ 4.8, (Delbarre *et al*, 1996)] depends on the environmental pH. Because of contrasted pH between cytoplasmic and apoplastic compartments

1 Department of Medical Genetics, Faculty of Biology and Medicine, University of Lausanne, Lausanne, Switzerland
2 Swiss Institute for Bioinformatics, Lausanne, Switzerland
3 Centre for Integrative Genomics, Faculty of Biology and Medicine, University of Lausanne, Lausanne, Switzerland
4 Laboratoire de Reproduction et Développement des Plantes, CNRS, INRA, ENS Lyon, UCBL, Université de Lyon, Lyon, France
 *Corresponding author. Tel: +41 21 692 5452; E-mail: sven.bergmann@unil.ch
 **Corresponding author. Tel: +41 21 692 3941; E-mail: christian.fankhauser@unil.ch
 †These authors contributed equally to this work
 ††These authors contributed equally to this work

(estimated at 7 and 5.5, respectively) (Kurkdjian & Guern, 1989; Bibikova *et al*, 1998; Yu *et al*, 2000; Kramer & Bennett, 2006; Krupinski & Jonsson, 2010), an auxin fraction can be passively imported by the cell, while only active transport allows for auxin export. Whether regulation of apoplastic pH is required for auxin gradient formation and phototropic bending, to our knowledge, has not been thoroughly investigated so far. Regulation and maintenance of the proton gradient across the plasma membrane and apoplastic pH requires the activity of plasma membrane-localized proton pumps of the AHA family (H⁺-ATPases) (Palmgren, 2001). H⁺-ATPase activity is crucial for a large variety of physiological processes such as stomatal opening, nutrient uptake, or hypocotyl and root growth (Palmgren, 2001; Haruta *et al*, 2010; Haruta & Sussman, 2012). Interestingly, the role of H⁺-ATPases has been linked to cell elongation by the acid growth theory (Rayle & Cleland, 1992; Hager, 2003; Cosgrove, 2005). It stipulates that cell elongation requires apoplastic acidification to activate cell wall-loosening proteins (Hager, 2003). Recently, it has been shown that auxin-induced cell elongation involves auxin-mediated regulation of H⁺-ATPase activity by phosphorylation (Takahashi *et al*, 2012; Spartz *et al*, 2014). Therefore, regulation of H⁺-ATPase activity might play a dual role during phototropism: to modulate the portion of protonated auxin and thus auxin influx, and to promote cell wall acidification and thus cell elongation.

To shed further light on auxin gradient formation during phototropism, we established an auxin flux model based on the morphology of the hypocotyl of an *Arabidopsis thaliana* seedling enabling us to test *in silico* the impact of various parameters: hypocotyl topology, apoplast thickness, and apoplastic pH changes. Our model predicted that regulation of apoplastic pH is a key step for the establishment of a lateral auxin gradient, a prediction that we supported experimentally. Finally, we provide results suggesting a mechanism explaining how light can regulate H⁺-ATPases and thereby potentially apoplastic pH at the molecular level.

# Results

### An *in silico* model for auxin flux during hypocotyl phototropism

Overall, auxin fluxes include active and passive cellular efflux and influx, and free auxin diffusion within the apoplastic compartment (Kramer, 2007; Krupinski & Jonsson, 2010). While the apoplastic diffusion distance depends on the actual apoplastic thickness and pH (Kramer, 2006), passive efflux and influx depend on compartmental pH and cell surface (Krupinski & Jonsson, 2010). Moreover, active fluxes are subject to carrier expression levels and localization.

To test *in silico* the impact of these various contributions on auxin gradient formation during phototropism, we used ordinary differential equations to build an auxin flux model. We considered active efflux contributions from both ABCBs and PINs (Supplementary Table S1), because members of both transporter families have been proposed to control auxin gradient formation upon phototropic stimulation (Christie *et al*, 2011; Ding *et al*, 2011; Haga & Sakai, 2012; Willige *et al*, 2013). We also explicitly considered fluxes resulting not only from passive influxes and effluxes in the cells but also from free diffusion in the apoplast.

Concerning active auxin transport, a starting modeling assumption in our model supported by experimental evidence is that upon unilateral blue light irradiation, PIN3 is polar in the endodermal cells on the lit side (Ding *et al*, 2011). In all other tissues, PINs and ABCBs are expressed apolarly. We did not consider active IAA (auxin indole-3-acetic acid) influx contributions resulting from AUX1/LAX for the following reasons: (i) A previous study showed that phototropism in the *aux1lax1lax3* triple mutant is not significantly different from the wild-type (Christie *et al*, 2011), (ii) this triple mutant lacks the expression of *AUX1* and *LAX3*, which were the most highly expressed members of the *AUX1/LAX* family in the hypocotyl (Supplementary Fig S1), and (iii) we observed that different double, triple, and the *aux1lax1lax2lax3* quadruple mutant showed a normal final phototropic response although in the quadruple mutant, there was a slight growth re-orientation delay (Supplementary Fig S1). Possible implications of including an AUX1/LAX term in our model are further evaluated in the discussion.

In etiolated *Arabidopsis* seedlings, light sensing occurs at the site of asymmetric growth, suggesting that formation of a lateral auxin gradient occurs locally (Iino, 2001; Preuten *et al*, 2013; Yamamoto *et al*, 2014). Thus, we assumed locality of gradient formation and used a realistic hypocotyl cross section to model gradient formation (Fig 1A and B). We tested the effect of a change in apoplastic pH, because small variations around the estimated resting apoplastic pH of 5.5 have a big impact on the protonation state of auxin influencing passive diffusion (the pKa of IAA is 4.8) (Kurkdjian & Guern, 1989; Bibikova *et al*, 1998; Yu *et al*, 2000; Kramer & Bennett, 2006; Krupinski & Jonsson, 2010) (Fig 1C). In all our simulations, pH modulation was treated as an exogenous variable, that is, a variable that is not affected by the model. pH modulation during phototropism therefore was imposed manually by modifying the apoplastic pH around cells on the shaded and/or lit side of the cross section. As we will discuss later on, a potential mechanism to create such a pH modulation is a light-triggered and phototropin-mediated modulation of H⁺-ATPase activity (see below).

Lastly, we took into account topological parameters like different apoplast thickness distributions as observed during seedling elongation (Derbyshire *et al*, 2007) as well as modifications of the available cell surface for the different cell layers (Fig 1D and E). The former was tested because apoplast thickness impacts auxin travel distances in the apoplast, while the latter is of interest since changes in size of the active interface between cells and apoplast modify absolute auxin flux contributions via a membrane. The cell surface variation was realized by modifying the classical cell size distribution of low-diameter endodermal and epidermal cells and large-diameter cortical cells and thereby affecting the relation between cell volume and cell surface (Fig 1D). We judged our models based on their ability to generate an auxin concentration difference in epidermal cells of opposing sides because the epidermis is considered as limiting for growth (Kutschera & Niklas, 2007; Savaldi-Goldstein *et al*, 2007) and auxin gradients were observed in the epidermis of photo-stimulated seedlings (Haga & Iino, 2006). To calculate concentrations, we used the cell and apoplast volumes and surfaces as obtained from a hypocotyl cross section (Fig 1D and E). We arbitrarily considered that a gradient was established when more than 1% concentration difference between opposing sides was obtained in the model.

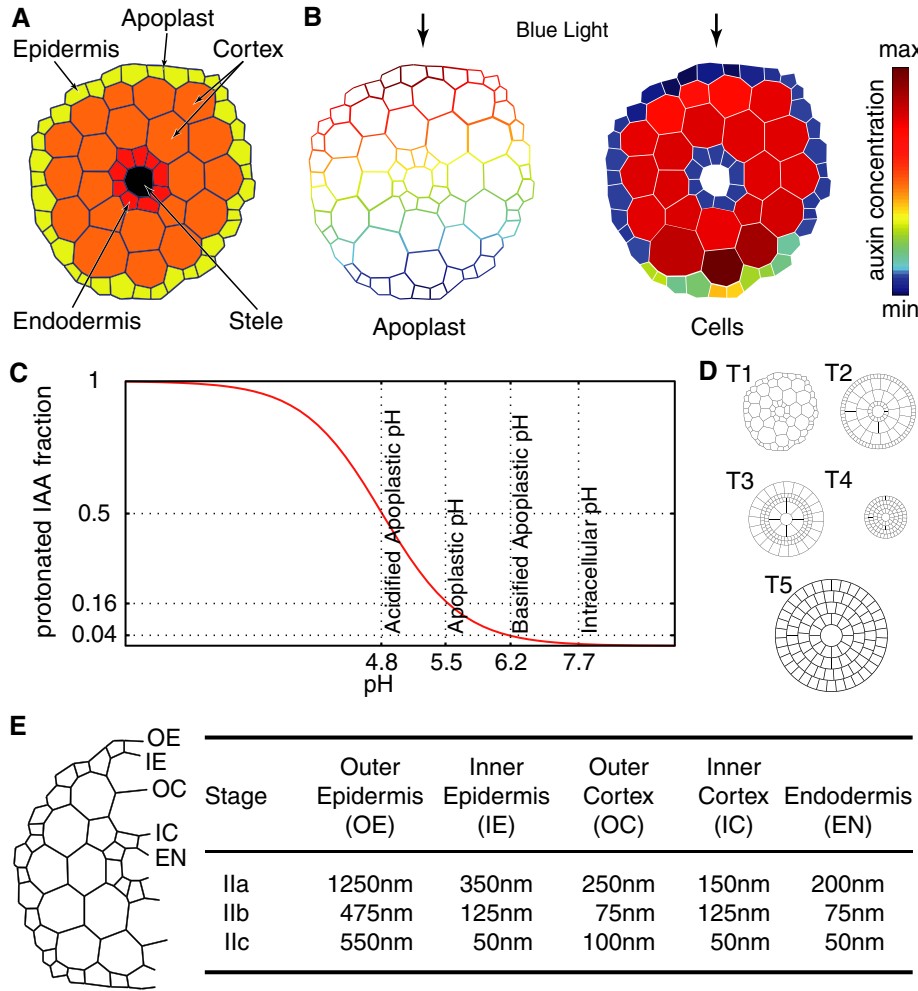

**Figure 1.  Overview on model domain, formed gradients, topological parameters, and biophysical parameters tested in the model.**

A  Modeled domain representing a cross section through the elongation zone of a 3-day-old etiolated *Arabidopsis thaliana* seedling.

B  Example of an auxin concentration gradient formed within a cross section showing apoplastic auxin gradient and cellular auxin gradient.

C  Dissociation curve for IAA based on its pKₐ of 4.8 showing protonated fractions for different compartmental pH values.

D  Different topologies tested during model parameter exploration: a realistic cross section (T1), a rotational symmetric cross section model with a cell size distribution over the different layers as found in the realistic cross section (T2), a rotational symmetric cross section model with an inverted cell size distribution (T3), and rotational symmetric cross section model where all cells have the same size (either small (T4)-like cells found in epi- and endodermis or big (T5)-like cells found in the cortex). Here, small cells have a diameter of approximately 15 μm, while big cells have a diameter of approximately 30 μm.

E  Illustration of the exact localization of the different apoplast layers, outer epidermis (OE), inner epidermis (IE), outer cortex (OC), inner cortex (IC), and endodermis (EN) and their measured thicknesses for different elongation states as reported by Derbyshire and colleagues (Derbyshire *et al*, 2007).

## Apoplastic pH modulation is necessary for auxin gradient formation

Among all variables tested, auxin gradient formation depended most critically on a modulation of the pH in the apoplast. Gradients could only be formed when the apoplastic pH around epidermal cells on the shaded side was lowered (Fig 2A; Supplementary Fig S2). Before acidification, we assumed an initial apoplastic pH of 5.5 (Kurkdjian & Guern, 1989; Bibikova *et al*, 1998; Yu *et al*, 2000; Kramer, 2006; Krupinski & Jonsson, 2010) and we assumed a drop in pH by 0.7 units. Such a drop in apoplastic pH has been observed previously (Fasano *et al*, 2001; Boonsirichai *et al*, 2003; Monshausen *et al*, 2011). Moreover, this

has a serious impact on the protonation state of the naturally occurring IAA: While at a resting pH of 5.5, only approximately 16% of the apoplastic auxin is protonated and is therefore able to permeate cell membranes, after acidification (to pH 4.8), the protonated fraction increases to 50%. Therefore, a drop in apoplastic pH creates a trap for apoplastic auxin and boosts the intracellular auxin concentration of surrounding cells. On the contrary, simulation of apoplast basification on the lit side was not sufficient to induce auxin gradient formation (Fig 2A). This might be explainable by the fact that increasing the apoplastic pH is only able to affect the protonation state of the approximately 16% of IAA already protonated (Fig 1C). We also modeled the effect of basification of the apoplast on the lit side

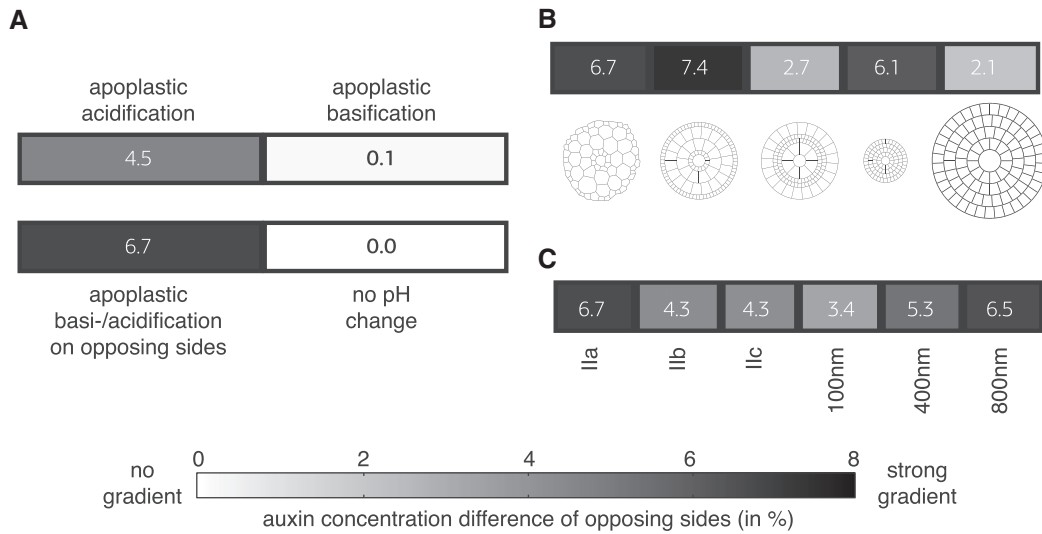

**Figure 2.  Impact of different parameters on *in silico* auxin gradient formation.**

As base scenario for a realistic cross section with apoplast thickness distribution IIa (corresponding to short cells), full PIN activity and concomitant acidification and basification were used.

A   Impact of modulations in apoplast pH distributions. Here, only the subset of scenarios in which we applied apoplast acidification shows lateral gradient formation.

B   Impact of different cell size distributions. Only the realistic, symmetrized realistic, and only-small-cells topologies are able to form lateral gradients.

C   Impact of apoplast thickness on gradient formation. Tested apoplast thickness distributions were distributions with relatively thick epidermal walls and thinner internal apoplast starting from very thick (IIa) ranging over medium (IIb) to small (IIc) and homogeneous apoplast thicknesses of 800, 400 and 100 nm, respectively. Only scenarios IIa, 400 and 800 nm are able to form lateral gradients.

while lowering the initial pH in etiolated seedlings from 5.5 to 4.8; however this did not increase the resulting gradients (Supplementary Fig S2). Finally, concomitant apoplast acidification and basification on opposing sides enhanced the gradients observed in the acidification-only scenario (Fig 2A).

**Topological parameters strongly modulate gradient formation**

Cellular topology has the potential to contribute to the formation of lateral auxin gradients for the following reasons. To result in an equivalent change in auxin concentration, more auxin molecules need to be transported in or out of a cell with a larger volume than a cell with a smaller volume. In addition, for the nearly cylindrical cells found in etiolated hypocotyls, the ratio of cell surface to cell volume decreases with increasing diameter. Considering that the cell surface is the interface via which auxin has to be moved, small diameter cells can change their auxin concentration more easily and faster. In addition, the cellular geometry impacts the apoplastic volume (see below).

We quantified the impact of hypocotyl morphology on gradient formation in our model. For calculations of auxin concentrations, we used cell and apoplast volumes and surfaces determined from the cross sections shown in Fig 1D. We found that topological features indeed have a strong impact on gradient formation. For simplification, we only considered cell size variations using idealized topologies (Fig 1D, topologies T2-4). When testing the impact of cell size distributions, we started with the natural cell size distribution with small cells (~15 μm in diameter) in epi- and endodermis and big cells (~30 μm in diameter) in the cortex (Fig 1D, topology T2) and tested inverted cell size distributions (Fig 1D, topology T3) as well as only small cells (Fig 1D, topology T4) and only big cells

(Fig 1D, topology T5). According to these simulations, the natural cell size distribution is beneficial for gradient formation (Fig 2B).

Notably, the idealized topology with realistic cell size distribution yielded a relatively similar gradient to the gradient simulated in the natural topology (Fig 2B), indicating that potential asymmetries in the realistic topology do not have a strong influence on gradient formation. To test this further, we also simulated gradient formation in a situation where light comes from a different side (rotated by 90 degrees) than in the original simulations. As our hypocotyl cross section and others that we found in the literature (Gendreau *et al*, 1997; Crowell *et al*, 2011) are not perfectly symmetric under (discrete) rotations, different directions could have led to different outcomes in our model prediction. Yet, our simulations showed that the observed differences were small (a few percent at most), indicating that asymmetries in our cross section do not affect our results significantly (Supplementary Fig S3).

In contrast, an inverted cell size distribution prevented gradient formation, as did a distribution consisting of only big cells. On the other hand, formation of an auxin gradient was possible using only small cells. In addition, we observed that further decreasing the cell volume while maintaining the cell surface constant further enhanced the steepness of gradients. This is a likely scenario in hypocotyl cells of etiolated seedlings that primarily consist of a large vacuole. Assuming that auxin is excluded from the vacuoles and that vacuoles in fully pressurized hypocotyl cells make up for at least 90% of the cell volume, simulations predict resulting gradients reaching up to 12% difference between shaded and lit side opposed to 8% when ignoring vacuoles and otherwise using the same settings. This corresponds to a 50% increase in gradient strength by considering potential effects of compartmentalization of the cells.

Apart from cell sizes and volumes, apoplast thickness also plays a potential role in auxin gradient formation. This is due to the fact that the apoplast potentially provides a mode of long-distance auxin transport depending on its diameter (Kramer, 2006, 2007). Considering that, depending on the elongation status and thus on the cellular geometry of hypocotyl cells their surrounding, apoplast thicknesses vary considerably (with apoplast thicknesses decreasing with increasing cell elongation) (Derbyshire *et al*, 2007), cell elongation and thickness might contribute to auxin gradient formation. To test this in our model, we compared thickness distributions documented during different elongation states of hypocotyl cells (Derbyshire *et al*, 2007). We particularly considered the early stages of hypocotyl elongation that all show relatively thick outer epidermal apoplasts and considerably thinner apoplasts on the inside. Using the thickness distributions reported by Derbyshire and colleagues (Derbyshire *et al*, 2007), we considered non-elongated cells (IIa), partly elongated cells (IIb), and strongly elongated cells (IIc) (Derbyshire *et al*, 2007) (Figs 2C and 1E). We contrasted these measured thicknesses with homogeneous thickness distributions within the range of those measurements (Derbyshire *et al*, 2007) (Fig 2C and Supplementary Fig S3).

The strongest gradients were found in scenarios using thick apoplasts (IIa, 800 nm, and a bit less in case of 400 nm) (Fig 2C). This supports the hypothesis that the apoplast constitutes an important mode of long-distance auxin flux during lateral gradient formation. Our analysis indicated that a thick epidermal layer was particularly favorable for gradient formation since thickness distribution IIa features a very thick outer epidermal apoplast (1,250 nm), but all the other layers are thinner than in the 400-nm scenario (Figs 1E and 2C). Despite thinner apoplast in the inner cell layers, gradients formed in scenario IIa were stronger than in case of the homogeneous 400-nm apoplast.

### Regulation of plasma membrane H⁺-ATPase activity is required for phototropism

Our *in silico* study predicted that the apoplast, in particular apoplastic pH regulation upon unilateral light perception, is a fundamental parameter for auxin gradient establishment (Fig 2A). Apoplastic pH is regulated by the activity of plasma membrane-localized H⁺-ATPases (Palmgren, 2001). To test this prediction experimentally, we first analyzed the kinetics of phototropic bending in conditions with an altered regulation of the apoplastic pH, by modulating the plasma membrane H⁺-ATPase activity. Treatment with increasing concentrations of the proton pumps inhibitor dicyclohexylcarbodiimide (DCCD) progressively inhibited the phototropic response in the wild-type (Fig 3A). Accordingly, we observed a delayed phototropic response of mutants lacking the expression of the two most expressed AHA proteins, *aha1* and *aha2* (Supplementary Fig S4). Inhibition of H⁺-ATPase activity by genetic or pharmacological approaches resulted in reduction of the hypocotyl growth rate (Supplementary Fig S5A and B), which could be the cause of reduced phototropism. However, we note that in a previous study, the growth rate of decapitated seedlings was similarly reduced as in seedlings treated with 50 μM DCCD, but decapitated seedlings still showed robust phototropism while DCCD-treated seedlings did not (Fig 3A) (Preuten *et al*, 2013). This indicates that reduced growth rate alone is not the reason for phototropism inhibition observed when H⁺-ATPase activity is reduced.

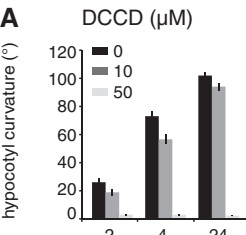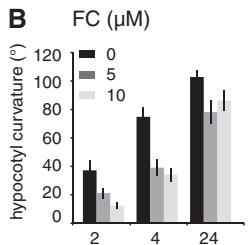

**Figure 3.  Regulation of H⁺-ATPase activity is required for optimal phototropism.**

A   Inhibition of proton pump activity by the proton pump inhibitor dicyclohexylcarbodiimide (DCCD) represses phototropism.

B   Enhancement of proton pump activity by fusicoccin (FC) treatment represses phototropism. Data represent the rate of hypocotyl growth curvature upon unilateral blue light irradiation with a fluence rate of 10 μmol m⁻² s⁻¹. Values are means ± 2×SE, *n* > 20.

Source data are available online for this figure.

To further investigate the role of H⁺-ATPases, we treated seedlings with the specific activator fusicoccin (FC). General activation of H⁺-ATPase activity upon FC treatment increased hypocotyl growth rate, but reduced phototropism (Fig 3B and Supplementary Fig S5C). The fact that both inhibition and activation of H⁺-ATPases during unilateral light treatment affect the amplitude of hypocotyl bending suggests that regulation of H⁺-ATPase activity is required for optimal phototropism.

### H⁺-ATPase regulation is necessary for auxin gradient establishment

We then tested whether H⁺-ATPase regulation is required for auxin gradient establishment during phototropism. We evaluated the distribution of auxin across the hypocotyl upon phototropic stimulation using the auxin sensor DII-Venus, a synthetic protein degraded directly upon auxin perception (Brunoud *et al*, 2012). The signal of DII-Venus was homogeneously distributed in the hypocotyl before phototropic stimulation. Following a 1-h unilateral blue light treatment, which precedes phototropic re-orientation in our conditions, we observed an asymmetric DII-Venus signal across the hypocotyl (Fig 4A and Supplementary Fig S6). The signal was stronger on the lit side compared to the shaded side, indicating a higher accumulation of auxin on the shaded side (Fig 4A and B). Treating seedlings with the auxin efflux inhibitor NPA completely abolished the formation of the gradient (Fig 4 and Supplementary Fig S6). Importantly, FC treatment strongly impaired auxin gradient establishment (Fig 4 and Supplementary Fig S6), demonstrating that misregulation of H⁺-ATPase activity and consequently misregulation of apoplastic pH during phototropic stimulation prevent auxin relocalization.

### Phototropins regulate plasma membrane H⁺-ATPase phosphorylation during phototropism

Since regulation of H⁺-ATPase activity is required for auxin gradient formation preceding phototropic bending, we examined how AHA proteins are regulated upon light perception in hypocotyls.

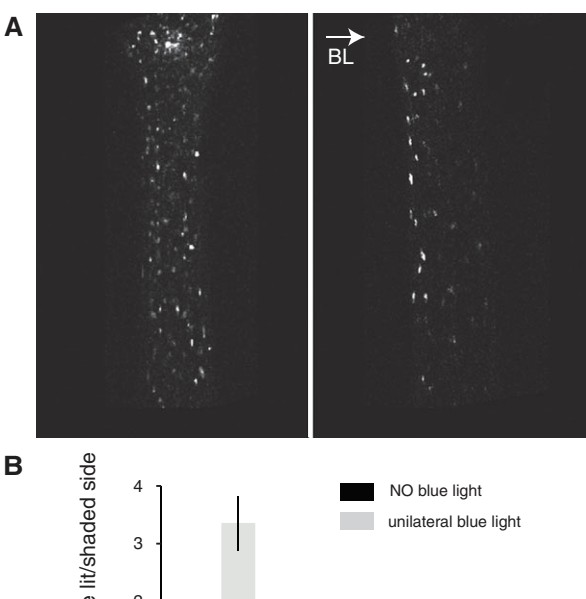

**B**

**Figure 4.**  Regulation of H⁺-ATPase activity is required for lateral auxin gradient formation.

A  DII-Venus signal in hypocotyls was examined before (left) or after (right) 1 h blue light (BL) irradiation with a fluence rate of 10 μmol m⁻² s⁻¹.

B  General activation of H⁺-ATPases and inhibition of auxin transport prevent auxin gradient formation. Seedlings treated with DMSO, NPA, or FC were analyzed as in (A). Quantification of DII-Venus signal was performed on 11–13 seedlings for each treatment, and data represent the ratio of the DII-Venus fluorescence signal between the lit side and the shaded side. Values are means, and error bars represent standard errors.

Source data are available online for this figure.

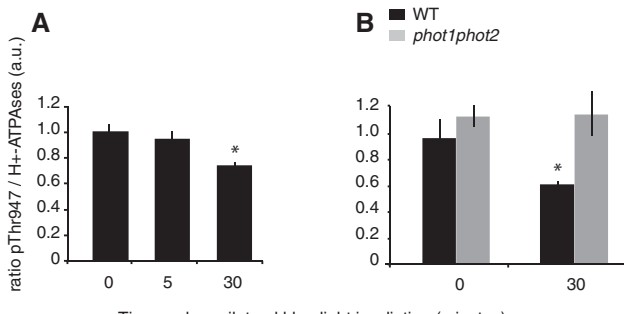

Time under unilateral blue light irradiation (minutes)

**Figure 5.**  Phototropins regulate H⁺-ATPase phosphorylation in hypocotyls upon light perception.

A  Unilateral blue light decreases H⁺-ATPase phosphorylation levels in hypocotyls. Three-day-old (WT) seedlings were either kept in darkness (0) or irradiated with 10 μmol m⁻² s⁻¹ unilateral blue light for the indicated time. Total proteins from dissected hypocotyls were separated by SDS–PAGE and transferred onto nitrocellulose membrane. Accumulation of total H⁺-ATPases and H⁺-ATPases phosphorylated at the penultimate amino acid was analyzed by immunoblotting using anti-H⁺-ATPases (H⁺-ATPases) and anti-phosphorylated-threonine-947 (pThr947) antibodies, respectively. Quantifications of pThr947 signal relative to the H⁺-ATPases total signal were performed on three biological replicates. Values are means, and error bars represent standard errors. * indicates significant difference between means of light-treated samples compared to dark control ($P < 0.05$).

B  Regulation of H⁺-ATPase phosphorylation levels at pThr947 depends on phototropins. Three-day-old seedlings of Col-0 (WT) or phot1phot2 mutant were either kept in darkness (0) or irradiated with 10 μmol m⁻² s⁻¹ unilateral blue light for 30 min. Proteins were analyzed as described in (A). *indicates significant difference between means of light-treated samples compared to dark control ($P < 0.05$).

Source data are available online for this figure.

for the need of specific regulation of H⁺-ATPase activity during phototropism and the stimulus-induced pH modulation predicted by our model.

## Discussion

We investigated the importance of different factors on lateral auxin gradient formation *in silico* by modeling auxin fluxes in an *Arabidopsis thaliana* hypocotyl cross section. We thereby assume locality of perception and response, which was recently demonstrated in *Arabidopsis* (Iino, 2001; Preuten *et al*, 2013; Yamamoto *et al*, 2014). The cross section used represents a natural topology including the apoplastic space surrounding the cells, which was explicitly represented because it provides a potentially important aspect of auxin transport (Kramer, 2007) and has also commonly been neglected in otherwise comparable models (Grieneisen *et al*, 2007; Wabnik *et al*, 2010; Santuari *et al*, 2011).

### Impact of morphological parameters on auxin gradient formation

Phototropic bending happens in the short hypocotyl cells in the elongation zone, and it is usually assumed that this is due to the lack of growth potential in elongated cells (Kami *et al*, 2012; Preuten *et al*, 2013; Yamamoto *et al*, 2014). Our results suggest that

Phosphorylation of H⁺-ATPases occurs at multiple sites and is an important mechanism regulating their activity (Duby & Boutry, 2009; Rudashevskaya *et al*, 2012). Phosphorylation at the penultimate residue, a threonine (Thr947 in *Arabidopsis* AHA2), is a primary step for the activation of H⁺-ATPases (Duby & Boutry, 2009). To evaluate the activity of H⁺-ATPases in hypocotyls upon light perception, H⁺-ATPases phosphorylation levels were analyzed by immunoblotting using an antibody recognizing the catalytic region of H⁺-ATPases and an antibody specifically recognizing the phosphorylated threonine (pThr947). These antibodies recognize several members of the AHA family (Hayashi *et al*, 2010). A decreased phosphorylation of H⁺-ATPases was detected in dissected hypocotyls when the seedlings were irradiated unilaterally with blue light (Fig 5A and Supplementary Fig S7). Importantly, while the level of H⁺-ATPase phosphorylation at the penultimate amino acid was similar between wild-type and the *phot1phot2* mutant in the dark, we did not observe any blue light regulation of H⁺-ATPase phosphorylation in the absence of phototropins (Fig 5B). Altogether, our data indicate that phototropins regulate H⁺-ATPase activity in the hypocotyl during phototropism. Consequently, these data provide a potential molecular explanation

in addition to a reduced ability to grow, elongated cells with concomitantly reduced apoplast thickness also have a reduced potential to form a lateral auxin gradient (Fig 2C and Supplementary Figs S2 and S3) (Derbyshire *et al*, 2007). During phototropic bending of the hypocotyl, the epidermis on the shaded side elongates the most of all layers (MacLeod *et al*, 1985; Orbovic & Poff, 1993) and it corresponds to the layer with the most wall material (Derbyshire *et al*, 2007). Thereby, the naturally observed topology favors both gradient formation and rapid elongation without the prior need for new cell wall synthesis.

Our model also predicts a strong impact of cell size distributions on gradient formation (Fig 2B and Supplementary Fig S2). Seedling morphology with small cells enhances the potential to form a lateral auxin gradient. Moreover, a layer of small epidermal cells, followed by large cortex cells and small endodermal cells, is favorable for the establishment of a lateral auxin gradient while inverting the sizes of the cells in the different cell types prevents gradient formation (Fig 2B). As expected for an embryonic organ, the hypocotyl cellular arrangement is stereotypical with 33–36 epidermal, 13 outer cortex, 8 inner cortex, and 8–9 endodermis cells (Fig 1) (Gendreau *et al*, 1997; Crowell *et al*, 2011). Intriguingly, the cell size distribution present in *Arabidopsis* hypocotyls appears to be conserved among angiosperms, suggesting the possibility of selection of morphological features that favor tropic hypocotyl growth (Meyer, 1971; Busse *et al*, 2005).

Our findings highlight the importance of cellular morphology that may constrain the formation of auxin gradients. Several recent studies have provided evidence for links between mechanical/cellular constraints and auxin-mediated growth processes (Heisler *et al*, 2010; Nakayama *et al*, 2012; Peret *et al*, 2012; Lindeboom *et al*, 2013; Lucas *et al*, 2013; Vermeer *et al*, 2014). Further investigating the relationship between cellular morphology, the associated biophysical constraints and auxin-mediated growth processes are important if we want to understand fundamental aspects of plant growth.

## A role for regulated H$^+$-ATPase activity in phototropism

We designed our starting modeling assumptions based on a paper that showed polar distribution of PIN3 in the endodermis of the lit side during phototropism (Ding *et al*, 2011). Our model suggests that polarization of PIN3 in the endodermis is not sufficient to create an auxin gradient (Fig 2 and Supplementary Fig S2). This does not mean that light-induced PIN3 relocalization is unimportant, but suggests that additional mechanisms are required to promote auxin gradient formation in photostimulated hypocotyls (see below). By testing a number of model parameters, we identified modulation of apoplastic pH as a key step to form a lateral auxin gradient across the hypocotyl of photo-stimulated seedlings. Apoplastic basification on the lit side and apoplastic acidification on the shaded side is the optimal combination to establish a lateral auxin gradient (Fig 2A). We propose that this leads to the concomitant increase and decrease in growth rates on opposing sides of the stem that has been documented previously (MacLeod *et al*, 1985; Orbovic & Poff, 1993). Interestingly, pH changes have been observed on the surface of gravi-stimulated *Arabidopsis* roots (Monshausen *et al*, 2011) and appear to be linked to intracellular pH modulation (Fasano *et al*, 2001; Boonsirichai *et al*, 2003;

Monshausen *et al*, 2011). Cytoplasmic basification (that correlates with extracellular acidification) occurs within 2 min of gravi-stimulation and precedes asymmetric distribution of PIN efflux carriers (Fasano *et al*, 2001; Boonsirichai *et al*, 2003; Monshausen *et al*, 2011).

We provide experimental evidence supporting this important modeling prediction; to modulate apoplastic pH, we interfered with the plasma membrane H$^+$-ATPase activity and showed that this disrupts formation of a lateral auxin gradient preceding phototropism (Fig 4B and Supplementary Fig S6). The fact that both H$^+$-ATPase activation and inhibition negatively influenced phototropic bending suggests that not only the proton pump activity but also an appropriate regulation of its activity is required for an optimal growth response during phototropic stimulation (Fig 3 and Supplementary Fig S6). We therefore propose that differential apoplastic pH regulation is achieved by a differential regulation of the H$^+$-ATPase activity on opposing hypocotyl sides, that is, activation on the shaded side and inhibition on the lit side.

Phototropins are the primary photoreceptors triggering phototropism, but how their activation leads to auxin gradient formation remains elusive (Sakai & Haga, 2012; Christie & Murphy, 2013; Hohm *et al*, 2013). Phototropin (phot1) interacts with and regulates the phosphorylation status of several proteins involved in phototropism (Pedmale & Liscum, 2007; Christie *et al*, 2011; Demarsy *et al*, 2012; Takemiya *et al*, 2013). Here, we demonstrated that phosphorylation of the plasma membrane H$^+$-ATPases in the hypocotyl is regulated by the phototropins (Fig 5B). We propose that the phototropin-mediated control of H$^+$-ATPase phosphorylation is important to establish asymmetric hypocotyl growth during phototropism. Asymmetric activation of the phototropins has been observed during unilateral seedling irradiation (Salomon *et al*, 1997); this may lead to differential phosphorylation/regulation of H$^+$-ATPases across a hypocotyl section.

We showed that in the context of hypocotyl phototropism, phototropin activation inhibits H$^+$-ATPase phosphorylation. Phototropin-mediated regulation of H$^+$-ATPase phosphorylation has been observed for other physiological responses. For example, during light-induced kidney bean movements, phototropin activation also leads to a dephosphorylation of H$^+$-ATPases in pulvini cells (Inoue *et al*, 2005). In contrast, in stomata, phototropin activation leads to enhanced phosphorylation of H$^+$-ATPases (Kinoshita & Shimazaki, 1999). In both cases, activation of phot1 leads to changes in the phosphorylation status of the penultimate threonine of AHA proteins, which regulates their activity. However, the steps leading from phototropin activation to the regulation of H$^+$-ATPases phosphorylation are not fully understood and depend on the context (Takemiya *et al*, 2013).

## Working model for lateral auxin gradient formation during phototropism

Our data highlight the importance of regulated H$^+$-ATPase activity in the establishment of an auxin gradient preceding phototropism (Figs 2–4). As auxin promotes H$^+$-ATPase activity (Chen *et al*, 2010; Takahashi *et al*, 2012), we propose a model that includes feedback and feed-forward loops between auxin transport and H$^+$-ATPase regulation, thereby promoting auxin gradient formation. On the lit side, reduced proton pump activity leads to apoplast basification

                                                                                    

decreasing auxin uptake, and in turn, the decrease in intracellular auxin further reduces proton pump activity. In contrast, on the shaded side, auxin uptake increases with H$^+$-ATPase activity leading to apoplast acidification. In turn, auxin accumulation in the cell further enhances proton pump activity. Thus, a network of interlaced regulatory loops controls auxin gradient formation during the phototropic response. Consistent with this idea, we show that interference with auxin transport (NPA) and with H$^+$ ATPase activity (FC) disrupts lateral auxin gradient formation (Fig 4 and Supplementary Fig S6). This reciprocal regulation of auxin concentration and H$^+$-ATPase activity is reminiscent of the complex relation between auxin concentration regulation and auxin transport as auxin regulates the expression and localization of its own transporters (Krecek *et al*, 2009).

Our mathematical model identified a novel mechanism required for auxin gradient formation that was validated experimentally (Figs 2–4). The strongest auxin gradients predicted by our model (12% by considering vacuolated cells) were lower than what was measured in maize coleoptiles and pea epicotyls, but nevertheless comparable to the 20% gradient determined in hypocotyls of *Brassica* which are closely related to *Arabidopsis* (Iino, 1992; Esmon *et al*, 2006; Haga & Iino, 2006). The relatively shallow gradient predicted by our simulation contrasts with the large difference in DII-Venus signal between the shaded and lit sides of the hypocotyl observed here (Fig 4). However, we do not know how the *in vivo* auxin concentration relates to the DII-Venus signal. Hence, it cannot be concluded that a threefold change in DII-Venus signal corresponds to a threefold change in auxin concentration. In our model, gradient strength is sensitive to auxin efflux carrier density, pumping capacity, and coupling of the modeled cross section to cross sections above and below not explicitly represented in the model (see Supplementary Materials and Methods). Thereby, the steepness of the gradient depends on these parameters. And while these parameters can have a strong effect on gradient strength, they do not impact the qualitative behavior of the model. We unfortunately lack precise measurements for these parameters; however, the sensitivity of our model to efflux carrier density and pumping capacity is in accordance with the experimental evidence showing that mutants lacking several PINs show delayed and reduced phototropic responses (Friml *et al*, 2002; Ding *et al*, 2011; Haga & Sakai, 2012; Willige *et al*, 2013). Additional factors that likely favor gradient formation are the newly identified plasmodesmatal gating mechanism (Han *et al*, 2014) and the activity of AUX1/LAX carriers (Band *et al*, 2014). Indeed, since AUX1/LAX carriers are proton symporters, one can hypothesize that they contribute to reinforcing the auxin gradient formation: Apoplastic acidification and increase in H$^+$ concentration on the shaded side potentially increase AUX1/LAX-mediated active auxin uptake on the shaded side (Lomax *et al*, 1995; Carrier *et al*, 2008; Steinacher *et al*, 2012). This could explain the delay in phototropism observed in the quadruple *aux1lax1-lax2lax3* mutant (Supplementary Fig S1). Importantly, a recent study has shown that within the root tip, members of the AUX1/LAX family are essential to determine which cells have high auxin levels (Band *et al*, 2014). Taken together with our results, we conclude that further studying of mechanisms controlling entry of auxin into cells is very important to understand the distribution of this hormone within plants. To extend our model and to refine our hypotheses, it would be interesting to include the contribution of the AUX1/LAX family and the feedbacks between auxin transport and pH regulation (Carrier *et al*, 2008; Krecek *et al*, 2009; Lomax *et al*; Steinacher *et al*, 2012). Finally, once the link between phototropin activation and H$^+$ ATPase activity is better understood, it could be included directly into the model [similarly to the implementation of auxin-induced apoplastic acidification described by Steinacher and colleagues (Steinacher *et al*, 2012)] instead of treating pH change as an exogenous variable.

## Materials and Methods

### Model

Description of the model, parameters and equations are provided as Supplementary Materials and Methods and our modeling software coded in Matlab is available as Supplementary Code.

### Plant material and growth conditions

The Columbia (Col-O) ecotype of *A. thaliana* was used as the WT. All the following transgenic line and mutant alleles were in the Col-O background: *aha1-6, aha2-4* (Haruta *et al*, 2010), *phot1-5phot2-1* (de Carbonnel *et al*, 2010), and DII-Venus (Brunoud *et al*, 2012). Seeds were surface-sterilized, sown on agar plates (½ strength MS pH 5.7 buffered with MES, 0.8% agar), and treated as described (Lariguet & Fankhauser, 2004). For pharmacological treatments, seeds were sown on nylon mesh (160 μm, Micropore) placed on the surface of the plate. Seedlings were grown for 3 days in darkness at 22°C before indicated treatment. Light intensities were determined with an International Light IL1400A photometer (Newburyport, MA) equipped with an SEL033 probe with appropriate light filters.

### Pharmacological treatments

Nylon meshes with 3-day-old etiolated seedlings were transferred 1 h before indicated light treatment onto freshly prepared plates supplemented by 0, 5, or 10 μM fusicoccin (FC, Sigma) and 0.01% DMSO, or 0, 10, or 50 μM dicyclohexylcarbodiimide (DCCD, Sigma) and 0.01% ethanol, or 10 μM 1-N-naphthylphthalamic acid (NPA, Duchéfa).

### Phototropism

Three-day-old etiolated seedlings (6- to 9-mm-long hypocotyls) grown on vertical agar plates were irradiated with 10 μmol m$^{-2}$ s$^{-1}$ unilateral blue light for 24 h. Pictures were taken with an infrared camera at different time points. Angles formed by the hypocotyl relative to vertical were measured with the NIH image software. Means, standard errors, and Student's *t*-test were performed on 50 seedlings minimum.

### DII-Venus signal visualization and quantification

Seedlings were grown as described for phototropism except that an additional 24-h treatment with white light (25 μmol m$^{-2}$ s$^{-1}$) was applied to induce de-etiolation. This treatment was necessary to allow detection of DII-Venus signal in the hypocotyl, and the

seedlings were still responding to phototropic stimulation and sensitive to FC treatment (Supplementary Fig S6B and C).

Imaging was performed on an LSM-510 laser-scanning confocal microscope (Zeiss). Serial optical sections were acquired and quantification was performed as described in Supplementary Fig S6A.

### Quantification of H⁺-ATPase phosphorylation level at Thr947

Three-day-old etiolated seedlings (6- to 9-mm-long hypocotyls) grown on vertical agar plates were irradiated with $10 \ \mu mol \ m^{-2} \ s^{-1}$ unilateral blue light or $10 \ \mu mol \ m^{-2} \ s^{-1}$ blue light from above for indicated time (0–30 min). Seedlings were fixed in EtOH-acetic acid solution (3:1) for 15 min and transferred into 75% EtOH for 1–3 h. Proteins were extracted from 25 hypocotyls sections grounded with a plastic pestles in 20 μl 1× phosphate-buffered saline (PBS) containing 6 M urea and overnight incubation at room temperature. After addition of 30 μl 2× Laemmli buffer, proteins (10 μl per lane) were separated on 9% SDS–polyacrylamide gels and transferred onto nitrocellulose with Tris-glycine buffer. The blots were probed with antibodies raised against the catalytic domain of AHA2, or antibodies that recognize peptide containing the phosphorylated Thr947 in AHA2 (Hayashi *et al*, 2010). These antibodies recognize not only AHA2 but also other H⁺-ATPase isoforms in *Arabidopsis* (Hayashi *et al*, 2010). Membranes were blocked in PBS, 0.1% Tween-20, and 5% nonfat milk (PBS-T-M) for 1 h at room temperature, incubated in presence of the primary antibodies overnight at 4°C, washed three times in PBS-T-M, incubated with a goat anti-rabbit IgG conjugated to horseradish peroxidase for 1 h at room temperature, and washed three times in PBS-T-M. Chemiluminescence signals were generated using Immobilon Western HRP Substrate (Millipore). Signals were captured with a Fujifilm ImageQuant LAS 4000 mini CCD camera system, and quantifications were performed with ImageQuant TL software (GE Healthcare) (Supplementary Fig S7).

**Supplementary information** for this article is available online: http://msb.embopress.org

### Acknowledgements

We thank Clémence Roggo who performed important preliminary experiments. We are grateful to the following colleagues who generously supplied us with important material: Toshinori Kinoshita (Nagoya University) for anti-AHA antibodies and providing advice on how to use them; Michael Sussman (University of Wisconsin) for *aha1* and *aha2* mutants; and Masao Tanaka (Nara Institute of Science and Technology) for the PIN3-GFP line. We thank Paula Rudall (Royal Botanical Gardens, Kew) for her advice and suggestions regarding hypocotyl morphology, Anupama Goyal and Mieke de Wit for their helpful comments on the manuscript, and Malcolm Bennett (University of Nottingham) for sharing unpublished information and fruitful discussions. This work was supported by the University of Lausanne, grants from SystemsX.ch "Plant growth in a changing environment" to C.F. and S.B. and the Swiss National Foundation (FNS 310030B_141181/1 to C.F).

### Author contributions

TH and ED designed and performed the research, analyzed data, and wrote the article. CQ, TP, and LA performed the research and analyzed the data. TV provided important material. SB and CF designed the research, analyzed the data, and wrote the article.

### Conflict of interest

The authors declare that they have no conflict of interest.

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
