## [Review Process File · Molecular Systems Biology]

Plasma membrane H⁺-ATPase regulation is required for auxin gradient formation preceding phototropic growth

Tim Hohm, Emilie Demarsy, Clément Quan, Laure Allenbach Petrolati, Tobias Preuten, Teva Vernoux, Sven Bergmann and Christian Fankhauser

Corresponding author: Christian Fankhauser, University of Lausanne

Review timeline:

Submission date:	02 March 2014
Editorial Decision:	27 April 2014
Revision received:	04 July 2014
Editorial Decision:	31 July 2014
Revision received:	20 August 2014
Accepted:	22 August 2014

Transaction Report:

Editor: Thomas Lemberger

1st Editorial Decision

27 April 2014

Thank you again for submitting your work to Molecular Systems Biology. We have now heard back from the two referees who agreed to evaluate your manuscript. As you will see from the reports below, the referees find the topic of your study of potential interest. They raise, however, substantial concerns on your work, which should be convincingly addressed in a major revision of this work.

The major points raised by the reviewers refer to the following issues:

- the assumption that the contribution of AUX/LAX can be omitted would need further experimental support (reviewer #1)
- the realism of the spatial model is also questionable (reviewer #1)

On a more editorial level, we would kindly ask you to provide:

- a machine-readable version of your model
- the key quantitative measurements that underly your analysis; these data can be provided either as 'dataset' files in supplementary information or as 'source data files' that are directly associated with specific figure panels (see also <http://msb.embopress.org/authorguide#a3.4.3>).

If you feel you can satisfactorily deal with these points and those listed by the referees, you may wish to submit a revised version of your manuscript. Please attach a covering letter giving details of the way in which you have handled each of the points raised by the referees. A revised manuscript will be once again subject to review and you probably understand that we can give you no guarantee at this stage that the eventual outcome will be favorable.

REFEREE REPORTS

Reviewer #1:

Adaptive responses to environmental signals in plants represent an area of great interest to many researchers. The corresponding author is an acknowledged expert studying plant responses to light. The current manuscript investigates how a gradient of the hormone auxin is able to form in response to a unidirectional light signal to trigger a phototropic growth response. The authors adopt a multicellular modelling approach to determine how an asymmetric auxin gradient may form to promote differential growth and cause hypocotyl bending towards the light source.

The authors initially describe the various parameters in their model. They argue only PIN and ABCB classes of auxin efflux carriers need to be included in their model based on experimental data and theoretical arguments. The experimental data is based on their observation that a quadruple mutant lacking all 4 AUX1/LAX genes does not exhibit a major defect in phototropism. Nevertheless, fig S1 clearly shows there is an effect. In addition, these genes are known to exhibit contrasting spatial expression patterns that might cancel each other's effects out, when all are knocked out. To rule this out the authors should provide information for single, double and triple aux1/lax mutant combinations and, ideally, describe their expression patterns (as this is currently poorly described for the hypocotyl tissue in the scientific literature). The theoretical data quoted is also questionable as it contradicts almost every other experimental and modelling study. We were interested to read that the authors refer to Steinacher et al. claiming that "the impact of influx carriers on auxin uptake to be at least one order of magnitude smaller than the impact of passive influx." However, having reread this reference, we were unable to find such a claim or any evidence to support it.

In the next section, the authors discuss the importance of apoplastic pH. With increasing apoplastic acidification, there is a greater proportion of protonated auxin, so a greater proportion of auxin can passively diffuse into the cells, and a smaller proportion can enter the cells via the influx carriers. However, AUX1 co-transport two protons with each anion of auxin and one may suppose that the higher concentration of H⁺ after acidification could result in more active influx (despite less anionic auxin); this would depend on which process dominates - mathematically such a flux would depend on anionic auxin concentration times H⁺ concentration squared, so with a smaller anionic auxin concentration and larger H⁺ concentration, the flux could become smaller or larger. Nevertheless, nearly all published models assume that there is plenty of H⁺ so the influx isn't limited by the level of H⁺. Hence, with apoplastic acidification the influx depends entirely on the reduction of the anionic auxin concentration. I recommend the authors read the recent paper by Band et al, 2014, *Plant Cell*, which provides a comprehensive theoretical study backed up by experimental validation.

Commendably in their models the authors attempted to use realistic cell shapes. Nevertheless, they are still idealised. Cell/Tissue templates based on multiple cross sections of real hypocotyls would be best as recently demonstrated by Peret et al (2013) in *MSB* and Band et al (2014) in *Plant Cell*. In the former case the authors greatly benefited from a reviewer making this point as it helped reveal that a network component PIN3 was necessary to provide robustness to auxin response patterns when faced with variation in cell and tissue geometries exhibited between samples. Nevertheless, the authors go on to report interesting relationships between cell size and auxin gradient formation similar to that reported by Kramer in *TIPS* in 2005. To their credit, they also consider the impact of vacuoles on auxin gradient formation. The role of sub cellular/cellular/tissue geometry in auxin transport models is poorly discussed, so this represents a valuable contribution to alert readers of its necessity and quantitative impact.

Next, the authors demonstrate that PM-H⁺ATPase activity is required for phototropism providing compelling lines of pharmacological, genetic and reporter-based lines of evidence. They then go onto to demonstrate that PHOT1/2 blue light receptors phosphorylate PM-H⁺ATPase in response to blue light. This represents a very interesting molecular mechanisms and underlines the importance of pH regulation.

In summary, a very interesting manuscript whose theoretical and experimental findings are likely to appeal to many readers of *MSB*.

Minor Points:

1. In this discussion, the authors write "the apoplastic space.... has also commonly been neglected in otherwise comparable models (Band et al, 2012; ...)." However, Band et al 2012 does not model auxin transport, so the model contains no apoplast to be neglected.
2. The equations stated in the Supplementary text neglect the influence of the membrane potential on the active flux terms. These are key as active transport is driven by the electrochemical gradient across the cell membrane. See for example, the factors $N(\phi)$ in the equations described in Appendix 1 of Heisler and Jonsson *J. Plant Growth Reg.* 25:302-312 (2006).
3. A single-cell model of the role of ATPases was presented by Steinacher et al. This should be referenced. How does the proposed model relate to that previously published?

Reviewer #2:

In this paper the authors present a simplified computer model of a thin slice of Arabidopsis hypocotyl. This is used to assess the relative importance of various features of the tissue geometry, pH changes, and auxin concentrations during the phototropic response.

Having inferred that an apoplastic pH gradient might regulate the auxin gradient, they experimentally confirmed that chemical inhibitors of the proton pumps reduce phototropic bending.

Main questions/comments:

The authors' model is unusual in that it tends to downplay the importance of PIN auxin efflux carriers in the formation of the auxin gradient that drives tropic bending. Rather, if I read the model correctly, the main source of the auxin gradient is an apoplastic pH gradient, with acidification on the shaded side of the hypocotyl.

The auxin gradient that results appears to be no greater than 8% side-to-side, according to figure 2 (although Figure 1 seems to show more pronounced differences?). Since their DII-Venus data suggest an auxin gradient of 3x or more (compare Fig. 4A with the dose-response curves in Fig. 2A of Band et al. *PNAS* 2012), I wonder whether the model is capturing the size of the gradient accurately. They should address this in the text.

The authors focus on a version of the model that doesn't seem very realistic. They consider a disk-like section of the hypocotyl, just one cell layer thick, bordered from above and below by cell walls. This is fine. However, they assume these transverse cell walls provide a constant source of auxin to the adjacent cells. The biological basis of this hypothesized auxin source is confusing to me. It would make more sense if the cells themselves were all synthesizing small amounts of auxin, so that cells were the source of auxin, rather than the cell walls.

Minor comments:

Supplemental table S1 needs a little more space between columns 2 and 3.

In the supplement, they write "In other words, this means that the model does not consider inherent means on how photo stimulation impacts pH."
Please clarify this sentence. What is "inherent means".

The supplement lists the auxin decay rate as 0.00075. What are the units of this number?

Why does the multiplier 4.7 appear in the supplemental table values listing C_{PGP} and C_{PIN} , and what are the units of these numbers?

On page 6, they cite Steinacher 2012 as reporting that the effect of active influx is an order of magnitude smaller than passive influx. I couldn't find this statement in the cited paper. Clarify?

We would like to thank you and the reviewers for their constructive comments. A detailed point-by-point response is attached to this letter. Briefly, we addressed the major points raised by the reviewers and yourself as follows:

1. Request for further experimental support for our assumption that the contribution of AUX/LAX can be omitted would need (reviewer #1):

In order to address this issue we performed phototropism experiments in additional aux/lax mutant combinations. Our data show normal phototropic bending in the *aux1lax1lax2lax3* quadruple mutant, 3 triple mutants combinations and 3 double mutant combinations (new Figure S1). These genetic data further reinforced the notion that for an initial version of the model it is reasonable to omit members from the AUX/LAX family. We also analyzed expression of AUX/LAX family members in the hypocotyl and found that AUX1 and LAX3 are most strongly expressed in this tissue. These data guided us in the selection of mutants that were analyzed for phototropism (new Figure S1).

2. Question on the realism of our spatial model (reviewer #1 and 2)

Our model is realistic as it was obtained from an Arabidopsis hypocotyl cross-section. In order to determine whether asymmetries in this cross-section have an influence on model predictions, we simulated the irradiation from a different angle (90 degree shifted to the original simulations). This in effect is similar to testing a different topology and had no consequence on the model prediction (Figure S3).

3. Editorial request to provide a machine-readable version of our model and data

We started our work on the model in 2010 during a time where the spatial modeling support of SBML was still in its early days and not yet sufficient for our model. We therefore decided to code it in MATLAB directly. We are sorry for this inconvenience but happily provide all our MATLAB source code such that others can use our model and replicate our simulations. We also provide all our experimental data in a single Excel document. Depending on your preference we can make the code and data available upon request, as supplementary material at MSB and/or on a dedicated page on our website.

Point-by-point response:

Reviewer #1:

The authors initially describe the various parameters in their model. They argue only PIN and ABCB classes of auxin efflux carriers need to be included in their model based on experimental data and theoretical arguments. The experimental data is based on their observation that a quadruple mutant lacking all 4 AUX1/LAX genes does not exhibit a major defect in phototropism. Nevertheless, fig S1 clearly shows there is an effect. In addition, these genes are known to exhibit contrasting spatial expression patterns that might cancel each other's effects out, when all are knocked out. To rule this out the authors should provide information for single, double and triple aux1/lax mutant combinations and, ideally, describe their expression patterns (as this is currently poorly described for the hypocotyl tissue in the scientific literature).

Our first comment about this remark is that by analyzing the quadruple *aux1lax1lax2lax3* mutant we look at a situation where all closely related members of this gene family are eliminated and thus we minimize possible compensation effects among members of the same gene family. Hence this experiment allows us to look at the phototropic response in the absence of AUX1/LAX-mediated activity. Indeed our experiments show that in the experimental conditions tested this mutant shows a slightly slower phototropic response but the final bending angle is not different from the WT. To address the reviewers' comment experimentally we have now analyzed phototropism in the *aux1lax1lax2lax3* quadruple mutant, 3 triple mutants and 3 double mutants. Our results show that all mutants reach a similar final bending angle (new Figure S1). These genetic data further reinforced the notion that for an initial version of the model it is reasonable to omit members from the AUX1/LAX family.

In order to obtain information about the expression of the different members of the AUX1/LAX family we dissected hypocotyls and determined expression from the 4 members of the gene family by RT-Q-PCR. These data are presented in the new Figure S1, showing that AUX1 and LAX3 are most strongly expressed in the hypocotyl.

The theoretical data quoted is also questionable as it contradicts almost every other experimental and modelling study. We were interested to read that the authors refer to Steinacher et al. claiming that "the impact of influx carriers on auxin uptake to be at least one order of magnitude smaller than the impact of passive influx." However, having reread this reference, we were unable to find such a claim or any evidence to support it.

We are referring to Figure 4A/B from this paper (we include this figure at the end of this document). The different subpanels show the contributions of passive influx (P) and AUX1/LAX dependent influx (A) under different assumption (panels A vs. panel B) for varying apoplastic IAA concentrations. Under most assumptions passive influx appears to be about one order of magnitude higher than the one of AUX1/LAX mediated influx. However, it is true that around a

concentration of about 10^{-5} , the contributions of A and P are quite similar. We have clarified this in the text.

In the next section, the authors discuss the importance of apoplastic pH. With increasing apoplastic acidification, there is a greater proportion of protonated auxin, so a greater proportion of auxin can passively diffuse into the cells, and a smaller proportion can enter the cells via the influx carriers. However, AUX1 co-transporters two protons with each anion of auxin and one may suppose that the higher concentration of H^+ after acidification could result in more active influx (despite less anionic auxin); this would depend on which process dominates - mathematically such a flux would depend on anionic auxin concentration times H^+ concentration squared, so with a smaller anionic auxin concentration and larger H^+ concentration, the flux could become smaller or larger. Nevertheless, nearly all published models assume that there is plenty of H^+ so the influx isn't limited by the level of H^+ . Hence, with apoplastic acidification the influx depends entirely on the reduction of the anionic auxin concentration. I recommend the authors read the recent paper by Band et al, 2014, Plant Cell, which provides a comprehensive theoretical study backed up by experimental validation.

We agree with this general comment and did not intend to contradict the work published by Band et al., 2014. In fact there is an interesting conceptual analogy between Band et al., 2014 and our work that we have explained more carefully in our revised manuscript. Band et al. conclude that LAX/AUX are needed to control which tissues have high auxin levels whereas the PINs control the direction of auxin transport in the tissue. We also conclude that which tissues/cells have high auxin levels depends on auxin influx but for reasons explained above we concentrated our analysis on the importance of passive influx (P) rather than active AUX/LAX-mediated influx (A). This is now included in the discussion "Importantly, a recent study has shown that within the root tip members of the AUX1/LAX family are essential to determine which cells have high auxin levels (Band et al, 2014). Taken together with our results we conclude that further studying of mechanisms controlling entry of auxin into cells is very important to understand the distribution of this hormone within plants. To extend our model and to refine our hypotheses it would be interesting to include the contribution of the AUX1/LAX family and the feedbacks between auxin transport and pH regulation (Carrier et al, 2008; Krecek et al, 2009; Lomax et al; Steinacher et al, 2012)."

Commendably in their models the authors attempted to use realistic cell shapes. Nevertheless, they are still idealised. Cell/Tissue templates based on multiple cross sections of real hypocotyls would be best as recently demonstrated by Peret et al (2013) in MSB and Band et al (2014) in Plant Cell. In the former case the authors greatly benefited from a reviewer making this point as it helped reveal that a network component PIN3 was necessary to provide robustness to auxin response patterns when faced with variation in cell and tissue geometries exhibited between samples.

We have used a similar approach to the one described by Band et al., 2014 (using microscopic data from confocal sections) to generate our cellular model. It is also useful to remember that obtaining a cellular model of a hypocotyl cross-section is more simple than obtaining a cellular model from a longitudinal root section (Band et al., 2014). Indeed, for our cellular model of a hypocotyl cross-section we make a section of cylindrical cells orthogonally to their longitudinal axis. Consequently, no matter at what height a cell is cut, it approximately has the same diameter. In contrast, longitudinal sections typically generate more variability in cell size unless the cutting plane is exactly through the central axis (which is difficult to predict), This has implications on the area or volume (depending if it is a 2D or a 3D model) of the cell represented in the model. It is therefore necessary to find a fitting plane (e.g. defined by Bezier curves).

Nevertheless, we agree that potential asymmetries in our topology can impact our simulation results. Therefore, we did additional simulations where we simulated the irradiation coming from a different angle (90 degree rotated from the original simulations). This in effect is akin to testing a different topology, but saved us from performing additional time-consuming imaging, which had not been feasible in the given time frame. This simulation gave consistent results with our original simulation (see New Figure S3). Together with our (already reported) findings for idealized, rotationally symmetric topologies this provides strong evidence that our results do not depend on the precise cellular topology. Finally, we wish to point out that we found a few hypocotyl cross-sections in the literature (Gendreau et al., 1997 in Plant Phys; Crowell et al., Plant Cell 2011) and noticed that they were very similar to ours (as one might expect for an embryonic organ). This information was included in the revised manuscript.

Minor Points:

1. In this discussion, the authors write "the apoplastic space... has also commonly been neglected in otherwise comparable models (Band et al, 2012; ...)." However, Band et al 2012 does not model auxin transport, so the model contains no apoplast to be neglected.

Sorry for this mistake, we removed this reference from this discussion point.

2. The equations stated in the Supplementary text neglect the influence of the membrane potential on the active flux terms. These are key as active transport is driven by the electrochemical gradient across the cell membrane. See for example, the factors $N(\phi)$ in the equations described in Appendix 1 of Heisler and Jonsson J. Plant Growth Reg. 25:302-312 (2006).

We extended the description in the Supplement to incorporate the fact that the equations depend on $N(\phi)$. It now reads as follows: "... is subject to a transport capacity/density for the respective transporter combined with a constant reflecting membrane potential effects on active transport related fluxes ($C_{\text{"PIN"}}$ or $C_{\text{"PGP"}}$) ...".

3. A single-cell model of the role of ATPases was presented by Steinacher et al. This should be referenced. How does the proposed model relate to that previously published?

We made sure to properly cite this paper. We also wish to point out that ATPase activity is not explicitly modeled. We clarified this in the manuscript and added in the discussion. "Finally, once the link between phototropin activation and H⁺ ATPase activity is better understood it could be included directly into the model (similarly to the implementation of auxin induced apoplastic acidification described by Steinacher and colleagues (Steinacher et al, 2012)) instead of treating pH change as an exogenous variable." Furthermore, we extended the supplementary information to reflect this as well: "In other words, this means that the model does not consider inherent means on how photo stimulation impacts pH, e.g. by the hypothesized photo stimulus dependent regulation of H⁺-ATPases or other yet uncovered means of regulation. Especially the hypothesis about a phototropin based regulation of H⁺-ATPase provides a possible link between phototropism and apoplastic pH and could be included in a future version of the model inspired by the implementation by Steinacher and colleagues for a link between H⁺-ATPase activity and pH (Steinacher et al. 2012)."

Reviewer #2:

The authors' model is unusual in that it tends to downplay the importance of PIN auxin efflux carriers in the formation of the auxin gradient that drives tropic bending. Rather, if I read the model correctly, the main source of the auxin gradient is an apoplastic pH gradient, with acidification on the shaded side of the hypocotyl.

This is indeed one of the surprising/interesting findings that we made. As it is very difficult in biology to demonstrate that something does not happen we have not concentrated our analysis on the role of PIN proteins but rather followed up on our model prediction regarding the importance of apoplasm acidification.

Interestingly, a recent paper by Band et al., 2014 in Plant Cell also showed that auxin-induced influx into cells is particularly important to determine which root cells have high auxin concentrations. This is now discussed in our revised manuscript "Importantly, a recent study has shown that within the root tip members of the AUX1/LAX family are essential to determine which cells have high auxin levels (Band et al., Plant Cell 2014). Taken together with our results we conclude that further studying mechanisms controlling entry of auxin into cells is very important to understand the distribution of this hormone within plants. To further extend our model and to refine our hypotheses it would be interesting to include the contribution of the AUX1/LAX family and the feedbacks between auxin transport and pH regulation (Carrier et al, 2008; Krecek et al, 2009; Lomax, 1995; Steinacher et al, 2012)."

The auxin gradient that results appears to be no greater than 8% side-to-side,

according to figure 2 (although Figure 1 seems to show more pronounced differences?). Since their DII-Venus data suggest an auxin gradient of 3x or more (compare Fig. 4A with the dose-response curves in Fig. 2A of Band et al. PNAS 2012), I wonder whether the model is capturing the size of the gradient accurately. They should address this in the text.

The strongest auxin gradients predicted by our model is 12% (by considering vacuolated cells). This is lower than what was measured in maize coleoptiles and pea epicotyls but remarkably comparable to the 20% gradient determined in hypocotyls of Brassica, which are closely related to Arabidopsis (Esmon et al, 2006; Haga & Iino, 2006; Iino, 1992). Moreover it is important to keep in mind that the relationship between DII-Venus signal and auxin concentration in the hypocotyl is not known and we therefore cannot assume that this relationship is linear. We therefore conclude that we cannot use DII-Venus data to make statements about the strength of the auxin gradient. We have amended the text to clarify this issue. In the discussion we write “The relatively shallow gradient predicted by our simulation contrasts with the large difference in DII-Venus signal between the shaded and lit sides of the hypocotyl observed here (Fig. 4). However, we do not know how the *in vivo* auxin concentration relates to the DII-Venus signal. Hence, it cannot be concluded that a three-fold change in DII-Venus signal corresponds to a three-fold change in auxin concentration.”

The authors focus on a version of the model that doesn't seem very realistic. They consider a disk-like section of the hypocotyl, just one cell layer thick, bordered from above and below by cell walls. This is fine. However, they assume these transverse cell walls provide a constant source of auxin to the adjacent cells. The biological basis of this hypothesized auxin source is confusing to me. It would make more sense if the cells themselves were all synthesizing small amounts of auxin, so that cells were the source of auxin, rather than the cell walls.

There currently isn't any direct evidence showing that auxin production within hypocotyl cells is required for a normal phototropic response. This explains why we decided not to include an auxin production term within hypocotyl cells. In contrast, there is ample evidence that auxin is mainly produced in leaves/cotyledons from where it is then transporter down into the hypocotyl through the vasculature, the apoplasm and from cell to cell. These known modes of transport are present in our model assumptions.

Following this remark we have tested the phototropic response in two auxin biosynthesis mutants that were previously shown to be defective for shade-induced auxin-dependent hypocotyl elongation: *sav3* and *yuc1yuc4* (Tao et al., 2008 in Cell; Won et al., 2011 in PNAS). We performed phototropism experiments and found that both mutants had a perfectly normal response further justifying our modeling assumption. We are happy to provide these data if the reviewer is interested.

Minor comments:

Supplemental table S1 needs a little more space between columns 2 and 3.

This was corrected.

In the supplement, they write "In other words, this means that the model does not consider inherent means on how photo stimulation impacts pH." Please clarify this sentence. What is "inherent means".

We clarified this in the manuscript and the respective part in the supplement now reads as follows: "In other words, this means that the model does not consider inherent means on how photo stimulation impacts pH, e.g. by the hypothesized photo stimulus dependent regulation of H⁺-ATPases or other yet uncovered means of regulation. Especially the hypothesis about a phototropin based regulation of H⁺-ATPase provides a possible link between phototropism and apoplastic pH and could be included in a future version of the model inspired by the implementation by Steinacher and colleagues for a link between H⁺-ATPase activity and pH (Steinacher et al. 2012)."

The supplement lists the auxin decay rate as 0.00075. What are the units of this number?

It is 1/s for simple degradation proportional to its concentration. We added it to Table S1 in the supplement.

Why does the multiplier 4.7 appear in the supplemental table values listing C_PGP and C_PIN, and what are the units of these numbers?

It is in molecules per surface unit. We added it to Table S1 in the supplement.

On page 6, they cite Steinacher 2012 as reporting that the effect of active influx is an order of magnitude smaller than passive influx. I couldn't find this statement in the cited paper. Clarify?

We are referring to Figure 4A/B from this paper (we include this figure at the end of this document). The different subpanels show the contributions of passive influx (P) and AUX1/LAX dependent influx (A) under different assumption (panels A vs. panel B) for varying apoplastic IAA concentrations. Under most assumptions passive influx appears to be about one order of magnitude higher than the one of AUX1/LAX mediated influx. However, it is true that around a concentration of about 10⁻⁵, the contributions of A and P are quite similar. We have clarified this in the text.

Superimposition made from figure 4a of Steinacher paper showing an overall higher contribution of Passive influx (P) compared to AUX1/Lax mediated Active influx in Auxin fluxes.

Thank you again for submitting your work to Molecular Systems Biology. We have now heard back from the two referees who agreed to evaluate your manuscript. As you will see from the reports below, referee #2 still raises several issues.

The major point is to use more realistic scenarios for the source and transport of auxin; this reviewer provides concrete suggestions in this regard. Inclusion of the data on the phototropism experiments on *sav3* and *yuc1yuc4* would also be helpful.

As you may know, we allow in principle only a single round of revision. We feel however that in this case, we can allow you to revise the study to convincingly address the last points raised by reviewer #2 in an exceptional last round of revision.

With regard to the availability of the model in a machine-readable format, we understand that the model was create when SBML would not have been able to represent spatial modeling. If there is a way to update the model and convert it to the most recent version of SBML (level 3), that would be ideal. Otherwise, we would ask you to add in Supplementary information a zip archive with the MATLAB scripts. Please include a README file at the top leve of the archive to explain the content of each file.

If you feel you can satisfactorily deal with these points and those listed by the referees, you may wish to submit a revised version of your manuscript. Please attach a covering letter giving details of the way in which you have handled each of the points raised by the referees.

REFeree REPORTS

Reviewer #1:

The authors have doen an excellent job addressing both reviewers comments and improving the manuscript considerably to a standard expected for publication in MSB.

Reviewer #2:

1 The authors have not adequately replied to my comment about the source of auxin assumed in their computer simulations. They assume the cell walls have a constant auxin concentration, while I had suggested that the auxin might be better modeled as synthesized within the cells.

In response, the authors claim to have tested (data is not provided) two auxin mutant lines with deficits in auxin synthesis, and observed no phototropism deficit. This is indeed fair evidence that auxin synthesis is not a major player in the hypocotyl. On the other hand, both TAA1/TAR and YUC are redundant gene families, so the authors tests do not rule out a role for local auxin biosynthesis.

If local auxin biosynthesis does not play a role, it instead suggests that the longitudinal transport of auxin from the shoot apex is relevant to this system. Indeed, the authors say in their rebuttal that auxin arrives in the hypocotyl by transport from the apex, down through the vascular cylinder, and then via the apoplast to the outer cell layers. This seems very reasonable.

However, this scenario does not match their model assumption, which does not correspond to any realistic scenario: They assume the auxin concentration in the apoplast is constant. This would correspond to an apoplast that can supply an arbitrarily large amount of auxin to a cell without itself becoming depleted, and with no time lag to allow for diffusion of a renewed auxin supply from the vascular cylinder.

There are a couple of model possibilities the authors might pursue that I might find more sensible than their current version: 1. They could limit the constant auxin source to the apoplast of the central vascular cylinder. As the location of auxin transport from the apex, this region will presumably have the highest sustained apoplastic auxin concentration and the fastest depletion response times. 2. They could build a version of their model with cytoplasm-localized auxin biosynthesis.

2 The fact that the model gradients are very small - the authors say 12% side-to-side is the maximum - is worrying. This gradient is too small to be reliably measured in plants, and I suspect it is too small for the plant to respond as well. It would be encouraging to know that the model they propose is capable of producing larger gradients, under reasonable parameter choices.

3 The citation to Steinacher et al. (2012) to support their claim that carrier-mediated influx is small compared to diffusive influx is still not justified, even considering the figure (Fig. 4) cited by the authors. Steinacher et al. does not make an effort to estimate realistic flux values for the auxin carriers. For example, their permeability value for active efflux is 0.14 mm/h, 40 times too small for auxin transport.

4 With reviewer #1, I would be curious to know how influx carriers are distributed in this tissue, and efflux carriers, too. Both would be required to make strong quantitative conclusions about this system. But this may be well beyond the scope of the authors' current paper, which is qualitative. As for reviewer #1 comments on the pH-dependence of the influx carrier, this does not require a theoretical analysis, as several measurements have been made on this (e.g. Yang et al. current biol. 2006, Fig 3A).

2nd Revision - authors' response

20 August 2014

Editor's comment

The major point is to use more realistic scenarios for the source and transport of auxin; this reviewer provides concrete suggestions in this regard. Inclusion of the data on the phototropism experiments on sav3 and yuc1yuc4 would also be helpful.

Our main revisions are listed here and further explained in detail in the response to the specific comments of reviewer 2.

We added the data on the phototropism experiments for the *sav3* and *yuc1yuc4* mutants (new Figure S8). These data are not supportive of model 2 of the reviewer, we have therefore not tested it further but discussed these experiments with the necessary caution. As pointed out by the reviewer, while our experiments do not provide evidence for a role of local biosynthesis, they do not demonstrate that auxin biosynthesis cannot occur locally.

We performed additional simulations to test the first model suggested by reviewer 2. Discussing these data is important because it allowed us to make important clarifications to the text (manuscript page 21, SI pages 2 and 3) which in its previous version led to some misunderstandings. Moreover, these data also pertain to comment 2 of reviewer 2.

In this revision we have now also included a zip archive with the MATLAB scripts including a README file, a visual summary image, a short summary of our main findings and 3 bullet points.

Reviewer 1

We thank reviewer 1 for his helpful comments throughout this reviewing process and are glad to hear that he is fully satisfied by our revision.

Reviewer 2

1 *The authors have not adequately replied to my comment about the source of auxin assumed in*

their computer simulations. They assume the cell walls have a constant auxin concentration, while I had suggested that the auxin might be better modeled as synthesized within the cells.

This comment seems to be born from a misunderstanding. In our model cell wall acidification in single wall compartments not just results in increased passive auxin uptake by adjacent cells but at the same time lowers the auxin concentration in the wall compartment. Nevertheless, assuming free diffusion of auxin in the cell wall allows auxin molecules from neighboring wall compartments to move into the auxin depleted compartment. The net-flux of auxin is determined by the concentration difference of adjacent wall compartments, the relative distance between centroids of adjacent wall compartments, the diffusion speed of auxin in the cell wall, and the area connecting adjacent wall compartments. It is represented by the first term in Equation (2) of the supplement and repeated here:

$$\sum_{i \in N_a(a)} \frac{[IAA_i] - [IAA_a]}{d(m_a, m_i)} D_{IAA} A_{a,i}$$

As documented by Figure 1B of the manuscript, the steady state apoplastic auxin concentration distribution shows a minimum around cells on the shaded side for which the apoplastic space is acidified. Considering that prior to acidification auxin in the apoplast was homogeneously distributed, this shows that apoplastic auxin concentrations are not constant.

The fact that we consider free auxin diffusion in the apoplast is clearly stated in the manuscript:

“We also explicitly considered fluxes resulting not only from passive in- and effluxes in the cells but also from free diffusion in the apoplast.”

In the main manuscript we only used the word ‘constant’ once referring to cell surfaces. We thus suspect that the misunderstanding might have come from the supplement when we stated “auxin concentrations in the apoplast just above and below are kept constant”. We will come back to this point below.

In response, the authors claim to have tested (data is not provided) two auxin mutant lines with deficits in auxin synthesis, and observed no phototropism deficit. This is indeed fair evidence that auxin synthesis is not a major player in the hypocotyl. On the other hand, both TAA1/TAR and YUC are redundant gene families, so the authors tests do not rule out a role for local auxin biosynthesis.

We totally agree with this comment of the reviewer and already stated in our previous reply concerning the effect of PINs that it is very difficult in biology to demonstrate that something does not happen. But as the reviewer agrees, we consider our additional experiments (the data of which we added as Fig S8) to be “fair evidence that auxin synthesis is not a major player in the hypocotyl.” In combination with the fact that our modeling results are rather qualitative, we think it is sufficient to show that auxin biosynthesis in the hypocotyl does not seem to have a major impact. Still, accounting for the fact that our additional experiments cannot rule out that local auxin biosynthesis plays a role *in planta*, we added a respective remark in the part of the supplement discussing auxin sources as well as referencing the data of our additional experiments:

“The model thereby neglects a potential contribution of modulated auxin biosynthesis within cells of the modeled cross section. This assumption seems to be confirmed by phototropism essays in two auxin biosynthesis mutants that were previously shown to be defective for shade-induced auxin-dependent hypocotyl elongation *sav3* and *yuc1yuc4* (Tao et al., 2008 in Cell; Won et al., 2011 in PNAS) but show normal phototropic responses (see Fig. S8). Nevertheless, considering the redundancy in the *TAA1/TAR* and *YUC* gene families, a contribution of local auxin biosynthesis cannot be ruled out completely.” (On pages 2-3 of the SI)

If local auxin biosynthesis does not play a role, it instead suggests that the longitudinal transport of auxin from the shoot apex is relevant to this system. Indeed, the authors say in their rebuttal that auxin arrives in the hypocotyl by transport from the apex, down through the vascular cylinder, and then via the apoplast to the outer cell layers. This seems very reasonable.

We agree with this comment and have now included the following text in the supplementary information. “In addition, we incorporate the fact that the stele is the major mode of basipetal auxin transport and therefore has to be considered an auxin source. Further incorporating the fact that auxin transport in the stele is faster than in the apoplast (Kramer 2006), for our model we assume

that auxin concentrations in the stele are high and kept constant.” (On page 2 of the SI).

However, this scenario does not match their model assumption, which does not correspond to any realistic scenario: They assume the auxin concentration in the apoplast is constant. This would correspond to an apoplast that can supply an arbitrarily large amount of auxin to a cell without itself becoming depleted, and with no time lag to allow for diffusion of a renewed auxin supply from the vascular cylinder.

This comment seems to be born from a misunderstanding. Our model does not assume constant auxin concentrations within the cell wall but explicitly models auxin fluxes between neighboring apoplast compartments and between the apoplast and cells. Fluxes between apoplast compartments are assumed to be based on free diffusion.

The misunderstanding appears to arise from our discussion of the auxin supply to the modeled hypocotyl cross-section. As auxin source we consider the stele in which we assume a high and constant auxin concentration. This seems to be justified considering the higher rate of transport in the stele, effectively avoiding the problem of lag times. In addition we consider the apoplast as a second potential source of auxin in the model. In our previous version we had the following description for this part: “Apart from the fluxes within the cross section, we explicitly allowed for exchange with the apoplastic space just above and below the considered cross section. Since gradient formation is assumed to happen locally (Ino 2001; Preuten et al. 2013), auxin concentrations in the apoplast just above and below are kept constant, thereby functioning as an auxin source or sink depending on auxin redistribution in the modeled cross section.”

We suspect that this passage might have been the source of the misunderstanding. We agree that considering a constant auxin concentration above and below the modeled cross section is probably unrealistic but this is a limitation of our 2d model (we are currently not in a position to make a 3d model of the entire seedling). However, following the reviewers’ suggestion (modeling suggestion 1, see below), we have now tested the consequences of this exchange between the modeled section and the sections just below and just above. We now state the following in the supplement on page 2: “Our simulations have shown that this exchange with surrounding cross sections has a cushioning effect on the strength of gradients predicted by our model: For example, setting the exchange between the modeled cross section and cross sections above and below to zero results in qualitatively similar results but stronger gradients (increasing gradient strength from 12% for vacuolated cells to 82%). And although this documents a considerable effect of coupling strength on gradient strength, the fact that the qualitative behavior in both extremes is similar lets us not further pursue this parameter.”

We thus consider our model where cells outside the cross section have no auxin gradient at all as the “worst case scenario”, since it is likely that several layers of cells will develop a gradient, making any gradient strength between 12% and 82% in the mid-layer plausible. Studying the auxin gradient across several cross sections is certainly interesting but beyond the scope of our 2d model, which only aimed to make semi-quantitative statements about the necessary conditions for the formation of a gradient.”

There are a couple of model possibilities the authors might pursue that I might find more sensible than their current version: 1. They could limit the constant auxin source to the apoplast of the central vascular cylinder. As the location of auxin transport from the apex, this region will presumably have the highest sustained apoplastic auxin concentration and the fastest depletion response times. 2. They could build a version of their model with cytoplasm-localized auxin biosynthesis.

As discussed above we have now tested the 1st suggestion from the reviewer by considering no influence of the layer just above and just below the modeled cross-section with the exception of an auxin supply from the vasculature (see our detailed explanation in the previous point).

We have also discussed the 2nd suggestion from the reviewer (cytoplasm-localized auxin biosynthesis). We now included the phototropism data for *sav3* and *yuc1yuc4* mutants (Figure S8). Nevertheless, as mentioned before, we acknowledge that it is very difficult to demonstrate that something does not happen in biology and we therefore discuss these data with the necessary caution by writing: “Nevertheless, considering the redundancy in the *TAA1/TAR* and *YUC* gene families, a contribution of local auxin biosynthesis cannot be ruled out completely.” (page 3 of the SI)

**2* The fact that the model gradients are very small - the authors say 12% side-to-side is the maximum - is worrying. This gradient is too small to be reliably measured in plants, and I suspect it is too small for the plant to respond as well. It would be encouraging to know that the model they propose is capable of producing larger gradients, under reasonable parameter choices.*

We agree with the reviewer that a 12% difference in auxin concentration appears to be small. Nevertheless, we wish to make the following remarks.

- 1) There currently is no experimental data for auxin gradient strength in the hypocotyl of photo-stimulated Arabidopsis. Brassica is the closest Arabidopsis relative for which an auxin gradient was measured following phototropic stimulation and the authors determined a 20% gradient, which is rather comparable to our model prediction (Esmon et al, 2006).
- 2) As discussed above our model is sensitive to the coupling effect between the modeled layer and the layer just above and just below. Following the reviewers suggestion we have now tested this influence and our model with a 12% gradient can be considered as a “worst case scenario”. In the likely situation where a gradient is formed in several cell layers this value would be larger (82% maximum with no coupling considered).
- 3) We unfortunately have no precise values for auxin carrier density and pumping capacity which both influence gradient strength. All three parameters allow significantly strengthening of the gradient while staying within a reasonable range.

As pointed out above the scope of our 2d model is not to make a precise prediction of the strength of a gradient that we currently can't measure experimentally but rather to make predictions about the necessary conditions for the formation of a gradient. We rephrased the section in the manuscript on page 21 to clarify on this effect:

“In our model gradient strength is sensitive to auxin efflux carrier density, pumping capacity, and coupling of the modeled cross section to cell layers above and below not explicitly represented in the model (see supporting information). Thereby the steepness of the gradient depends on these parameters. And while these parameters can have a strong effect on gradient strength, they do not impact the qualitative behavior of the model. We unfortunately lack precise measurements for these parameters, however the sensitivity of our model to efflux carrier density and pumping capacity is in accordance with the experimental evidence showing that mutants lacking several PINs show delayed and reduced phototropic responses (Ding et al, 2011; Friml et al, 2002; Haga & Sakai, 2012; Willige et al, 2013).”

**3* The citation to Steinacher et al. (2012) to support their claim that carrier-mediated influx is small compared to diffusive influx is still not justified, even considering the figure (Fig. 4) cited by the authors. Steinacher et al. does not make an effort to estimate realistic flux values for the auxin carriers. For example, their permeability value for active efflux is 0.14 mm/h, 40 times too small for auxin transport.*

Citing Steinacher et al. in this context is not essential given that we have strong genetic support for our modeling assumption. Since the reviewer considers the evidence from Steinacher et al. as a weak argument for neglecting members of the AUX1/LAX family in our model, we removed the citation from the text.

**4* With reviewer #1, I would be curious to know how influx carriers are distributed in this tissue, and efflux carriers, too. Both would be required to make strong quantitative conclusions about this system. But this may be well beyond the scope of the authors' current paper, which is qualitative. As for reviewer #1 comments on the pH-dependence of the influx carrier, this does not require a theoretical analysis, as several measurements have been made on this (e.g. Yang et al. current biol. 2006, Fig 3A).*

We agree with the reviewer that measurements for carriers in cell/cell layer resolution would be extremely useful data and indeed are required for a strong quantitative conclusion. We further agree with the reviewer that this is beyond the scope of this study in which our argumentation is rather qualitative than quantitative.